# CONNECTIVITY-CONSTRAINED INTERACTIVE ANNOTATIONS FOR PANOPTIC SEGMENTATION

## ABSTRACT

Large-scale ground truth data sets are of crucial importance for deep learning based segmentation models, but annotating per-pixel masks is prohibitively time consuming. In this paper, we investigate interactive graph-based segmentation algorithms that enforce connectivity. To be more precise, we introduce an instance-aware heuristic of a discrete Potts model, and a class-aware Integer Linear Programming (ILP) formulation that ensures global optimum. Both algorithms can take RGB, or utilize the feature maps from any DCNN, whether trained on the target dataset or not, as input. We present competitive semantic (and panoptic) segmentation results on the PASCAL VOC 2012 and Cityscapes dataset given initial scribbles. We also demonstrate that our interactive approach can reach $90.6\%$ mIoU on VOC validation set with an overhead of just 3 correction scribbles. They are thus suitable for interactive annotation on new or existing datasets, or can be used inside any weakly supervised learning framework on new datasets.

## 1 INTRODUCTION

Deep Convolutional Neural Networks (DCNNs) excel at a wide range of image recognition tasks (He et al., 2016; Ren et al., 2017; Shelhamer et al., 2017), such as semantic segmentation (Chen et al., 2018b; Shelhamer et al., 2017; Zhao et al., 2017) and panoptic segmentation (Kirillov et al., 2019b; Yang et al., 2019; Xiong et al., 2019; Kirillov et al., 2019a). Semantic segmentation studies the tasks of assigning a class label to each pixel of an image, while instance segmentation (Chen et al., 2018a) detects and segment each object instance. Panoptic segmentation unifies both tasks that investigate to segment both *things* (such as person, cars) and stuff (such as road, sky) classes.

While DCNNs show outstanding results for semantic and panoptic segmentation, they have two conceptional problems. First, they require huge amounts of annotated data. Annotating image segmentation masks is a very time consuming and labor extensive task. For example, annotating a semantic image mask took "more than 1.5h on average" on the Cityscapes dataset (Cordts et al., 2016). In addition, DCNNs rely on their implicitly learned generalization probability and most of the state-of-the-art architectures do not make use of any domain specific knowledge, such as neighborhood relations and connectivity priors for segments. On the contrary, classical graph based segmentation models (Boykov et al., 2001; Yedidia et al., 2003) do not require any learning data and can incorporate specific domain knowledge. Their major drawback is that they rely on human-designed similarity features and require complex optimization algorithms or solvers, which are mainly CPU based and non-suitable for real time applications.

In this work, we explore the combination of DCNNs and graph-based algorithms for annotating ground truth segmentation. We propose two interactive algorithms that both enforce the *connectivity prior* (to be more precise in Sec. 2.1). Specifically, we design a heuristic region growing method based on the Potts model (Potts, 1952), as well as an integer linear programming (ILP) formulation of the markov random field (MRF) (Boykov et al., 1998). We utilize scribble based annotations from human annotators as initialized or iterative hard constraints for our algorithms, which is typical in a human-in-the-loop (HITL) annotation process. We explore two different scenarios. In the first scenario, we assume that for the segmentation task, there is already a pre-trained DCNN with the same class mapping. Thus, we could make full use of DCNN's probability map and scribbles as input to our algorithms. In the second scenario, we assume that no pre-trained DCNN for the same objective is available. This is true for a lot of existing datasets, e.g., Cityscapes does not contain any

Figure 1: Left: image with scribbles from Pascal VOC dataset. Mid-left: semantic segmentation result of our heuristic with layer 3 of ResNet 101. Mid-right: result of our ILP with probability map of DeepLab V2. Right: ILP without connectivity prior (with DeepLab V2 prob. map).

class labels for lane marking, which is crucial information for an autonomous vehicle. In this case, we cannot use the class specific probability map, but the more generic low level features of a DCNN can be utilized as feature description for the algorithm.

To investigate the general purpose of low level DCNN features, we compare DCNN trained on ImageNet (Russakovsky et al., 2015) and COCO (Lin et al., 2014) with and without fine tuning on the target dataset. Our experiments show that incorporating the connectivity prior as well as the DCNN features greatly improves the algorithm performance. We present competitive semantic (and panoptic) segmentation results on the PASCAL VOC 2012 (Everingham et al., 2015) and Cityscapes dataset. To the best of our knowledge, our method is the first "non-DCNN" panoptic segmentation algorithm on Cityscapes with competing results, which shows the potential improvement gained by a combined approach. See Fig. 1 for a visualization that uses initialized scribbles as input.

We also prototype our algorithms for the task of interactive annotation. Based on initial scribbles and a baseline DCNN's probability map, our algorithms acheive $90.9\%$ mIoU on VOC validation set, with just 3 rounds (1 scribbles per image each round) of correction scribbles.

Summarized, our key contributions are

- a novel heuristic algorithm and a class-aware ILP formulation with connectivity prior,

- an in depth analysis of a combination of DCNN and graph-based segmentation algorithms,

- extensive evaluation of the scribble based interactive algorithms for semantic and panoptic segmentations on two challenging datasets.

Our proposed algorithms have multiple use cases in annotating datasets for segmentation. First, they can be used inside any HITL annotation tool, as the annotator interacts with the image in forms of scribbles until satisfaction. Second, given initial scribbles, they can be used inside any weakly or semi-supervised learning framework for semantic (and panoptic) segmentation.

**Related Work.** The procedure of annotating per pixel segmentation masks is similar to interactive image segmentation, which is widely studied in the past decade. The method using bounding boxes (Rother et al., 2004) is suitable for instance segmentation, which requires the user to draw the box as tight as possible. Recently, 4 extreme points (Maninis et al., 2018; Papadopoulos et al., 2017) clicking is used as an alternative which shows superior result. On the other hand, polygon based methods (Russell et al., 2008) require users to carefully click the extreme points of things and stuff, and the accuracy heavily depends on the number of clicks. Among others, scribbles are recognized as a more user-friendly way (Boykov & Jolly, 2001; Levin et al., 2008). Moreover, it is also natural to annotate stuff classes using scribbles.

Modern annotation tools often adopt interactive deep learning based methods, including Polygon-RNN++ (Acuna et al., 2018) and Curve-GCN (Ling et al., 2019), which allows the annotator to click on the boundaries. Notably, a recent approach (Agustsson et al., 2019) enables both 4 extreme points clicking and scribbles correction. Besides, they can also take advantage of ensemble learning (Zhou, 2009), to combine several inference results to produce a better segmentation. However, these methods requires an existing ground truth dataset for DCNNs to learn on the first hand, which may not be available when unknown domains or new classes are introduced.

As a cheaper alternative, weakly supervised learning has drawn a lot of attention recently. (Dai et al., 2015; Khoreva et al., 2017; Li et al., 2018) claim that weakly iteratively trained by just bounding boxes and image tags, the DCNN can achieve $95\%$ segmentation score compared to fully supervised on VOC. Since this method emphasizes on thing classes, it has worse score (or even none) on stuff classes. Instead, (Lin et al., 2016; Can et al., 2018) claim that iteratively training a DCNN by scribble annotations suffers only a small degradation in performance on both thing and stuff classes.

For graph-based methods, the (discrete) Potts model (Potts, 1952) is widely used for denoising and segmentation. The authors of (Shen et al., 2017) formulate the problem as an ILP and try to solve the global optimum, but only to a reduced image size, while (Nguyen & Brown, 2015) proposed an efficient region-fusion-based heuristic algorithm. The MAP-MRF (maximizing a posterior in an MRF) has been well studied for image segmentation. Previous methods focus on local priors (Boykov & Veksler, 2006; Krähenbühl & Koltun, 2011), and efficient approximate algorithms exist, e.g., graph cut (Boykov et al., 2001) and belief propagation (Yedidia et al., 2003). MRF with connectivity prior has been studied in (Rempfler et al., 2016; Nowozin & Lampert, 2009) and formulated as an ILP, which can be solved by any ILP solver (Anand et al., 2017). Since solving an ILP exactly is in general $\mathcal{NP}$-hard (Land & Doig, 1960), they either solve a relaxation of the ILP, or focus on the binary MRF case.

Our proposed interactive (in forms of scribble) multi-label annotation algorithms are graph based, with global connectivity prior, i.e., they enforce pixels of the same label to be connected, which allows the annotator to better control over the final segmentation.

## 2 PROPOSED APPROACH

### 2.1 PREREQUISITE

Given an image, we build an undirected graph $G = (V, E)$ where the node set $V$ represents a set of pixels (or superpixels) and $E$ a set of edges consisting of un-ordered pairs of nodes. Image segmentation can be transformed into a graph labeling problem, where the label set $C$ is pre-defined.

When talking about segmentation, we need to first distinguish between class, instance, and region ID of a node. In semantic segmentation, the task is to assign a class label to each node in a graph. In panoptic segmentation, one has to further assign an instance ID to the node that belong to the "thing" class. In this paper, our algorithms require an additional region ID, which is linked to a scribble and we assume nodes with the same region ID must be connected (to be explained in Sec. 3.2). This is to deal with the case where an object of the 'thing' or "stuff" class is separated into several connected regions, e.g., the car in Fig. 2 is separated into two regions by a tree. Afterwards, the class and region labels can be used to generate a panoptic segmentation.

### 2.2 CONNECTIVITY-CONSTRAINED OPTIMIZATION ALGORITHMS

We first discuss the formal definition of connectivity, and the two proposed algorithms in details, i.e., the class-agnostic heuristic algorithm of the discrete Potts model and the class-aware ILP of MRF with connectivity constraints.

#### 2.2.1 THE CONNECTIVITY PRIOR

Two nodes $u, v$ in a graph $G$ are *connected* if there is a $(u, v)$-path in $G$. $G$ is called *connected* if every pair of nodes are connected in $G$, otherwise it is *disconnected*. Let $\bar{G}_\ell \subseteq G$ be a connected subgraph where every node is labeled $\ell \in C$. Then, the image segmentation with connectivity constrains corresponds to find a partition of $G$ into $k$ ($k = |C|$) connected (and disjoint) subgraphs. Enforcing connectivity constraints itself is proven to be $\mathcal{NP}$-hard in (Nowozin & Lampert, 2009).

#### 2.2.2 THE $\ell_0$ REGION FUSION BASED HEURISTIC

Given a graph $G(V, E)$, let $y_i$ be the information (either RGB or features from any DCNN) of node $i$, and $w_i$ be its estimated value, the discrete Potts model (Potts, 1952) has the following form:

$$\min_{\mathbf{w}} \sum_{i \in V} \|w_i - y_i\|_2 + \sum_{(i,j) \in E} \lambda \|w_i - w_j\|_0, \tag{1}$$

where $\lambda$ is the regularization parameter. Here, the first term is the data fitting and the second is the regularization term. We recall that the $\ell_0$ norm of a vector gives its number of nonzero entries.

In this paper, we introduce an iterative scribble based region fusion heuristic algorithm (which we call $\ell_0 H$) with the "class" and "region" ID for each node. In the beginning, the nodes covered by the same scribble are grouped together and labeled with the same IDs, while all other nodes are unlabeled and in their individual group. Note that different regions can share the same class ID. Then, the algorithm iterates over each group (outer loop) and its neighbors (inner loop) and decide whether to merge the neighbors or not, by checking their region IDs. If both groups have region IDs and are different, they cannot be merged. In all other cases, i.e., if both groups have no region ID or only one has it, the following merging criteria (Nguyen & Brown, 2015) are checked:

$$\sigma_i \cdot \sigma_j \cdot \|Y_i - Y_j\|_2 \leq \beta \cdot \gamma_{ij} \cdot (\sigma_i + \sigma_j). \tag{2}$$

where $\sigma_i$ denotes the number of pixels in group $i$, $Y_i$ the mean of image information (e.g., RGB color) of group $i$, and $\gamma_{ij}$ denotes the number of neighboring pixels between groups $i$ and $j$. Here, $\beta$ is the regularization parameter, and it increases over the outer loop number.

If (2) is satisfied, two groups are merged, and their labels are updated according to the following rule. If both groups have no "region" ID, the merged group still have none, hence unlabeled. If only one group has region ID, the merged group inherits the label, hence labeled.

After one outer loop over all groups, $\beta$ will increase and follows the exponential growing strategy of (Nguyen & Brown, 2015), i.e., $\beta = (\frac{\text{iter}}{50})^{2.2} * \eta$, where "iter" is the current number of outer loop and $\eta$ is the growing parameter. The procedure goes on until all groups are labeled, and the complexity of $\ell_0 H$ is $\mathcal{O}(n)$, where $n$ is the number of nodes.

Note that the above algorithm is approximate to problem (1), and connectivity of each region is enforced at every step. Given desired scribbles, $\ell_0 H$ is able to generate panoptic segmentations (and also semantic segmentations). Also note that the class ID does not play any role in the algorithm, it inherits from the scribble and propagates with the region ID. Hence, this algorithm is class-agnostic.

### 2.2.3 THE ILP FORMULATION WITH CONNECTIVITY CONSTRAINTS

The MRF with pairwise data term can be formulated as the following ILP:

$$\min_x \quad \sum_{\ell \in C} \sum_{i \in V} c_i^\ell x_i^\ell + \lambda \sum_{\ell \in C} \sum_{(i,j) \in E} d_{ij} |x_i^\ell - x_j^\ell| \tag{3}$$

$$\sum_{\ell \in C} x_i^\ell = 1, \quad \forall i \in V, \tag{3a}$$

$$x_i^\ell \in \{0, 1\}, \quad \forall i \in V, \quad \ell \in C, \tag{3b}$$

where $c_i^\ell$ denotes the unary data term for assigning class label $\ell$ to node $i$ (hence class-aware), $d_{ij}$ the simplified pairwise term for assigning $i, j$ different labels, and $\lambda$ is the regularization parameter. Constraint (3a) enforces that to each node is assigned exactly one label, i.e., $x_i^\ell = 1$ if and only if node $i$ is labeled $\ell$. Note that the absolute term can be easily transformed into linear terms by introducing additional continuous variables. The model above does, however, not guarantee connectivity, which is instead enforced as follows.

**Connectivity constraints with root node**  Let $r$ (the first node of scribble $\ell$) denote the root node of subgraph $\bar{G}_\ell$. Then, the following constraints of (Rempfler et al., 2016) suffice to characterize the set of all connected subgraphs of label $\ell$ that contain $r$

$$x_i^\ell \leq \sum_{s \in S} x_s^\ell, \quad \forall i \in V \backslash r : (i, r) \notin E, \quad \forall S \in \mathcal{S}(i, r), \tag{4}$$

where $S$ is the *vertex-separator set* of $\{i, r\}$, i.e., if the removal of $S$ from $G$ disconnects $i$ and $r$. And $\mathcal{S}(i, r)$ is the collection of all vertex-separator sets of $\{i, r\}$.

The number of constraints (4) is exponential with respect to the number of nodes in $G$, hence it is not possible to include into problem (3) all of them at in once. In practice, they are added iteratively when needed (called the cutting planes method (Kelley, 1960)). In this paper, we adopt the *K-nearest cut generation* strategy (Rempfler et al., 2016) to add a few of them at each iteration. See appendix A.1 for more details.

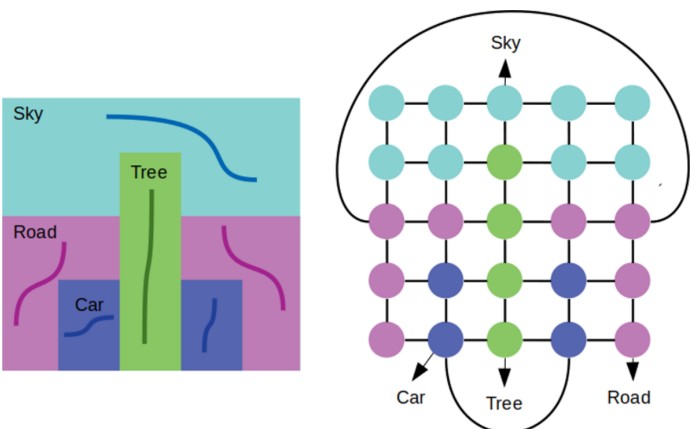

Figure 2: Left: our scribble policy to draw as many scribbles as there are separated regions. Right: ILP-P that introduces pseudo edges which connect multiple regions of the same class.

For the simple case where the region ID coincides with the class ID, i.e., the number of regions equals that of classes, problem (3) with connectivity prior is solved as follows. We first solve (3) and check if all subgraphs $G_i$ are connected. If not, we identify disconnected components, adds constraints (4), and solve the resulting ILP again. This procedure continues until all subgraphs are connected. This method ensures global optimumality if no time limit is restricted.

For the case where $k$ regions share the same class ID, we introduce an improved formulation (we call it ILP-P) where we add $k-1$ pseudo edges that "connect" all separated regions of the same class (illustrated in Fig. 2). In particular, this does not increase the number of variables and is class-aware. Then, the solving process follows the aforementioned framework. But ILP-P is only designed for semantic segmentations, i.e., it is not region-aware. Post-processing methods are needed to separate instances within the same class for panoptic segmentation.

## 3 OVERALL WORKFLOW

In this section, we describe the pre-processing steps as well as the scribble policy for our panoptic and interactive segmentation.

### 3.1 SUPERPIXELS AS DIMENSION REDUCTION

Superpixels have long been used for image segmentation (Stutz et al., 2017; Achanta et al., 2012), as they can greatly reduce the problem size while not sacrificing much of the accuracy. In this paper, we adopt SEEDS (den Bergh et al., 2012) to generate superpixels on the PASCAL VOC 2012 dataset, while using a deep learning based method (Tu et al., 2018) on the more challenging Cityscapes dataset. We then build a region adjacency graph (RAG) $G(V, E)$ of the superpixels, where each superpixel forms a node (vertex) and edges connect two adjacent superpixels.

### 3.2 EXTRACTING IMAGE FEATURES USING SCRIBBLES

Our segmentation algorithms are scribble supervised, which are two folded. On the one hand, the node labels, such as class, instance and region IDs are fixed if the nodes are covered by the corresponding scribbles. On the other hand, for the ILP algorithm, if no high level image information (i.e., probability map) exists, the scribbled superpixels will be used to extract information for the class, i.e., one can use the average of the scribbled superpixels information to represent each class.

Although the input to the algorithms can be as simple as image RGB information, one can also take advantage of the modern DCNN to extract deeper features. We distinguish two scenarios:

- No previous training data is available – one starts annotating images in a new dataset.

- Training data available – one continues annotating more images of an existing dataset.

In the former case, other than RGB, one can also adopt any base network (i.e., ResNet 101 (He et al., 2016)) pre-trained on other datasets (i.e., ImageNet) and use the output of the low level features that extracts image edges, textures, etc. In the later case, one can fully utilize any modern DCNN trained on the existing dataset, and use the output of the final layer (i.e., probability map).

**Scribble generation policy**   For panoptic segmentation, first of all, the scribble itself must be connected. Second, one has to draw as many scribbles as there are connected regions (both "thing" and "stuff" class) presented in the image. For example, if an object is cut into separated regions, one has to draw a scribble on each region. One sample image with scribbles is shown in Fig. 2.

### 3.3   INTERACTIVE SEGMENTATION USING SCRIBBLES

In a HITL annotation framework, it is typical that the annotator makes corrections to the current segmentation until satisfaction. In this paper, we simulate the human correction scribbles (described in Appendix A.3). We then report analysis of the segmentation results up to 3 rounds of scribbles correction in Sec. 4.4.

## 4   SEMANTIC AND PANOPTIC SEGMENTATION EXPERIMENTS

### 4.1   EXPERIMENTAL SETUP

In this session, we conduct extensive experiments on the Pascal VOC 2012 and Cityscapes validation set. In all our experiments, when we mention base network, we refer to the publicly released ResNet 101 that is pre-trained on ImageNet and COCO dataset. We adopt DeepLab V2 (Chen et al., 2018b) (without CRF as post-processing) and DRN (Yu et al., 2017) as our baseline DCNN to get the probability maps, trained on their corresponding training sets. We adopt IBM Cplex (Bliek et al., 2014) version 12.8 to solve the ILP. All computational experiments are performed on a Intel(R) Xeon(R) CPU E5-2620 v4 machine, with 64 GB memory.

We report the semantic and panoptic segmentation scores, where the pixel mean intersection over union (mIoU) is commonly used for semantic segmentation, and the panoptic quality (PQ) metric is newly introduce in (Kirillov et al., 2019b) and is a combination of segmentation quality (SQ) and recognition quality (RQ). The parameters for both heuristic and ILP is described in Appendix A.4.

### 4.2   RESULTS ON PASCAL VOC 2012

Pascal VOC 2012 has 20 "thing" classes and a single "background" class for all other classes. We evaluate our algorithms on the 1449 validation images. We first apply (den Bergh et al., 2012) to produce around 700 superpixels, and use the public scribbles set provided by (Lin et al., 2016) as initial scribbles. Since some of these scribbles do not meet our policy (described in Sec. 3.2), only the semantic segmentation is reported, in terms of mIoU averaged across the 21 classes. Visualization of 2 experiments are presented in Appendix A.5.

**No training data**   In this case, one can either use RGB or the output of lower level features of a base network as input ($y_i$) to our algorithms. We compare our class-agnostic heuristic ($\ell_0 H$) using RGB or different low level features from ResNet 101 as input, against ILP-P. We do not set any time or step limit for the heuristic, but a time limit of 10 seconds for ILP-P (denoted ILP-P-10). We report in Table 1 the detailed comparison, where we use the RGB, first and third layer of ResNet 101 as input to $\ell_0 H$, and "Dim" is the dimension of the input feature map.

We can see in Table 1 the advantage by incorporating lower level features maps of ResNet 101, that improves $\ell_0 H$ of RGB by 1.8%, even though it is pre-trained on completely different dataset. ILP-P adopts $\ell_0 H$-layer 3 as initial solution, and further increase the mIoU by 0.3%.

**With training data**   If training data is available, we use the probability map of DeepLab V2 (with baseline mIoU 70.5%) trained on VOC training set as input to our algorithms, $\ell_0 H$ get another huge boost of 10.1% to 81.7% compared to using layer 3. We compare ILP (3) without connectivity

Table 1: Comparison on VOC 2012 *val* set when no training data is available.

| Model | Dim | Time | mIoU |
|---|---|---|---|
| $\ell_0 H$-RGB | 3 | 2.2 | 69.8 |
| $\ell_0 H$-layer 1 | 64 | 2.9 | 70.8 |
| $\ell_0 H$-layer 3 | 256 | 3.9 | 71.6 |
| ILP-P-10 | – | 9.7 | **71.9** |

Table 2: Comparison when training data available, baseline DeepLab V2 (Chen et al., 2018b).

| Model | Time (s) | mIoU (%) |
|---|---|---|
| DeepLab V2 | – | 70.5 |
| MRF-prob | 0.2 | 80.8 |
| $\ell_0 H$-prob | 0.8 | 81.7 |
| ILP-P-10 | 7.9 | **84.6** |

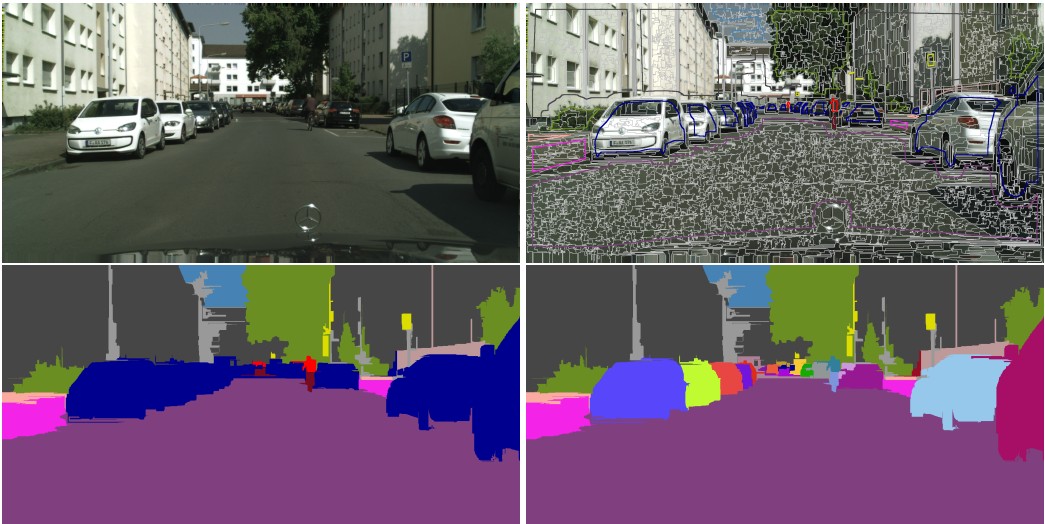

Figure 3: Top: Cityscapes original image and image with superpixels and artificial scribbles, Bottom: semantic and panoptic segmentation of $\ell_0 H$-prob, with DCNN's probability map as input.

that is solved to optimum (MRF-prob) with $\ell_0 H$ and found out it is $0.9\%$ worse, which shows the importance of the connectivity prior. We then compare ILP-P with $\ell_0 H$ as initial solutions ($81.7\%$ mIoU) and a time limit of 10 seconds, Table 2 suggests that ILP-P-10 improves the baseline of DeepLab V2 by $14.1\%$ and $\ell_0 H$-prob by $2.9\%$. An example of visual comparison is illustrated in Fig. 1. Since ILP (3) is class-aware and encodes pairwise term $d_{ij}$, further boost on the performance can be expected, given better baseline DCNN and an edge detector.

## 4.3 RESULTS ON CITYSCAPES

Cityscapes has 8 "thing" classes, and 11 "stuff" classes. While there is no public scribbles set, we simulate our own that meet the policy in Sec. 3.2. The procedure is described in Appendix A.2, and one can see Fig. 3 for an example. We evaluate our algorithms on the 500 validation images with the scribbles, where we first apply (Tu et al., 2018) to produce around 2000 superpixels.

**No training data** We use RGB and output of layer 3 of Resnet 101 as input to $\ell_0 H$, and report in Table 3 both mIoU and PQ. We also compare $\ell_0 H$ to one recent weakly supervised learning method (Li et al., 2018). It uses a more powerful PSPNet (Zhao et al., 2017) supervised by bounding boxes and image tags, but requires end to end training. $\ell_0 H$ shows superior results on both semantic and panoptic segmentation, both around $10\%$ boost compared to (Li et al., 2018). Note that it is not a fair comparison since ours requires scribbles while (Li et al., 2018) does not at inference time.

**With training data** We use the public full-supervised (trained on Cityscapes) DRN (Yu et al., 2017) as our baseline ($71.4\%$ mIoU), and run $\ell_0 H$ and ILP-P using its probability map. Table 3 shows $\ell_0 H$-prob improves the baseline by $4.4\%$, and by $1.5\%$ compared to $\ell_0 H$-layer 3 . Besides, PQ also increases from $49.6\%$ to $51.2\%$. Because of the rich class and instance information (more than 15 classes per image) of Cityscapes and also that we use around 2000 superpixels (around 30k

Table 3: Comparison on Cityscapes *val* set with and without its training data set. The later is based on DRN (Yu et al., 2017)'s probability map.

| Model | Time | mIoU | PQ | SQ | RQ |
|---|---|---|---|---|---|
| $\ell_0 H$-RGB | 6.5 | 74.2 | **49.6** | 74.3 | 63.8 |
| $\ell_0 H$-layer 3 | 7.2 | **74.3** | **49.6** | 74.5 | 63.7 |
| Weakly (Li et al., 2018) | – | 63.6 | 40.5 | – | – |
| DRN (baseline) | – | 71.4 | – | – | – |
| $\ell_0 H$-prob | 7.2 | **75.8** | **51.2** | 75.6 | 64.8 |

binary variables), ILP struggles to find any better solution within the time limit of 10 seconds. The score remains unchanged compared to $\ell_0 H$, and hence is not shown in the table. More visualization examples can be seen in Appendix A.6.

### 4.4 RESULTS ON INTERACTIVE SEGMENTATION

As described in Appendix A.3, we simulate 3 rounds of scribbles (1 scribble per image per round) to correct the largest error of previous segmentation. We report their mIoU, compared to the baseline DCNN (DeepLab V2) and current reported state-of-the-art (SOTA) deep learning approach (DeepLab V3+ with $84.6\%$) (Chen et al., 2018c) on VOC validation set. Results show that our algorithms benefit from addition scribbles and the performance increases significantly. The best result is reported by ILP-P-10 at iteration 3 ($90.6\%$), a $6.0\%$ mIoU gain with just 3 correction scribbles.

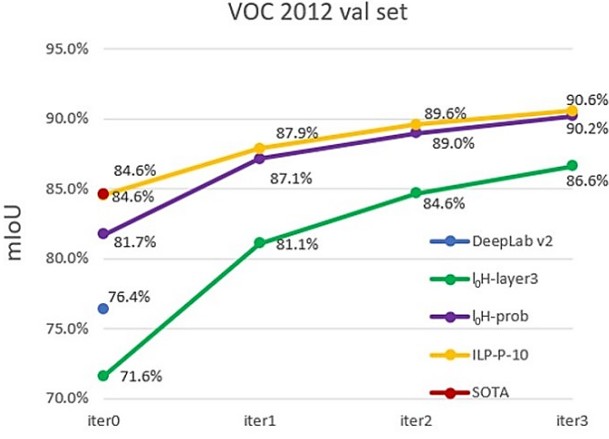

Figure 4: Interactive segmentation on VOC 2012 *val* set. The mIoU increases significantly with just one correction scribble per iteration. Blue dot is our baseline and red dot the current SOTA DCNN.

## 5 CONCLUSIONS AND FUTURE WORK

In this paper, we proposed two interactive multi-label graph based algorithms with connectivity prior. With DCNN's features and initial scribbles as input, both achieve competitive semantic (and panoptic) segmentation results on VOC and Cityscapes validation set. The interactive approach further boost the performance with little overhead of correction scribbles. It would be interesting to investigate our approach in a weakly supervised setting, where connectivity constraints are imposed on the outputs of a deep network to leverage unlabeled data. Training with such discrete high-order constraints can be explored via ADM (Alternating Direction Method) schemes, for instance, in way similar to Marin et al. (2019), which showed promising performances of imposing discrete MRF on the outputs of convolutional networks. Finally, note that all the reported performance are upper bounded by the accuracy of superpixels. Hence, using better superpixel algorithms or increasing its number may further influence the performance.

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

# A APPENDIX

## A.1 $K$-NEAREST CUT GENERATION ALGORITHM

We restrict ourselves to label $\ell$ only, and the approach described below will be repeated for the other labels. Upon solving problem (3), we denote those with $x^\ell = 1$ as active nodes, and we look for all connected components in $G_\ell$. We then iterate over all active components pair $(H, r)$, and use the following strategy to select $K$ constraints (4) each time.

Precisely, we run a breath-first search algorithm starting from $H$ to collect $K$ disjoint sets $S_m$ ($m = 1, \ldots, K$) where $S_m$ contains all the inactive nodes with distance $m$ to $H$. The search terminates if $K$ equals the number of nodes in $H$ or if another active node is reached. The idea is illustrated in Fig. 5, where active nodes are shown in black and $r$ denotes the root node. The two separator sets are marked in red and blue. Here $K = 2$, because it reaches the number of nodes in $H$. The $K$-Nearest strategy is reported in (Rempfler et al., 2016) to be one of the most successful (among five) in terms of solved instances and computational efficiency.

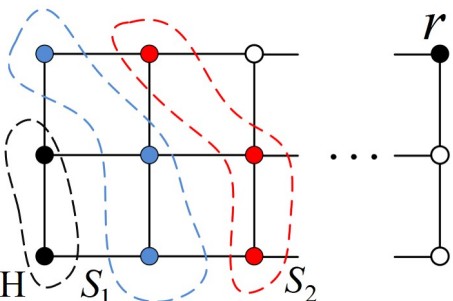

Figure 5: $K$-Nearest cut generation strategy. Active nodes are shown in black, and the two vertex separator sets are marked in red and blue.

## A.2 SIMULATING SCRIBBLE GENERATIONS

Since there is no public scribble set for Cityscapes, we hack the ground truth instance segment and apply erosion and skeleton algorithms to simulate scribbles that meet our policy in Sec. 3.2.

We generated artificial scribbles for both training and validation sets of Cityscapes. We adopt kernel size adapted erosion and skeleton method and apply them on the instance-level ground truth masks of Cityscapes. Therefore, the format of the scribbles are also the same as Cityscapes, which is 16 bit grayscale PNG images with both class and instance labels. The code that generates the scribbles will be open-source. All scribbles can be downloaded online [1].

## A.3 SIMULATING CORRECTION SCRIBBLES

For VOC 2012, we simulate only one scribble per image per iteration. Given the VOC ground truth and current segmentation mask, we first identify the largest error region which is connected and mislabelled. To simulate scribble from the error region, we first apply opening morphological operation to smooth the region. Then, we use skeletonization to shrink the region to 1 pixel wide without breaking the connectivity. We can then get the scribble by cutting branches in the skeleton tree. One can see Fig. 6 for an illustration of generating correction scribbles. The code will also be open-source soon.

## A.4 PARAMETERS SETTING FOR THE EXPERIMENTS

Parameter $y_i$ is the information contained in pixel $i$, which could come from a vector of RGB channels, layer 1 or layer 3 feature of ResNet 101, or from the probability map of a trained DCNN.

---

[1]https://drive.google.com/file/d/17sE1DDvqMDzdkpSNtrR7ulFxtfxDJ49P/view?usp=sharing

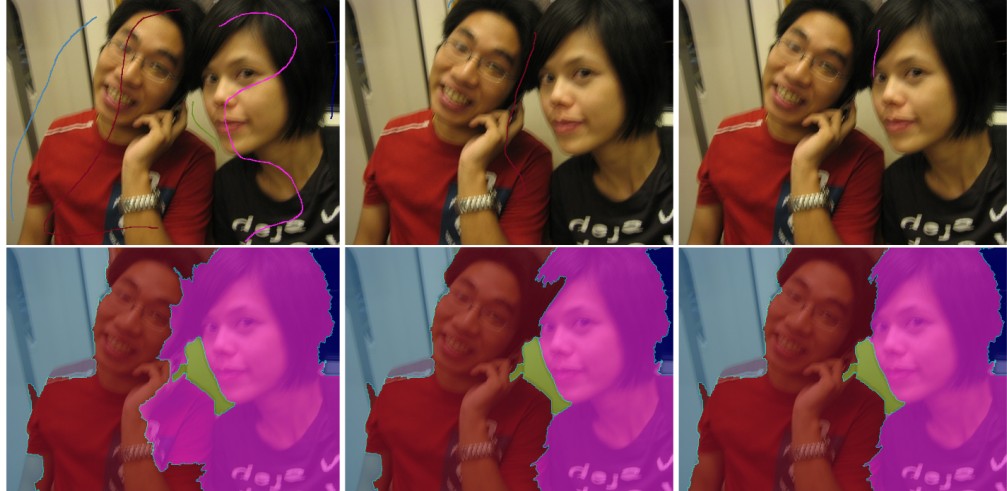

Figure 6: Top: Images from VOC with initial scribbles and 2 rounds of scribbles correction. Bottom: Their corresponding panoptic segmentation result.

For VOC 2012, the probability map comes from a trained DeepLab V2. For Cityscapes, we use DRN to get the probability map inference.

For $\ell_0 H$, after several trials, the growing parameter $\eta$ is set to 0.1, 20, 100 and 0.3 for RGB, layer 1, layer 3 and probability map of both VOC 2012 and cityscapes.

For ILP, when training data is available, we can use the probability map $p_i$ and $c_i^\ell = \left\| \mathbb{1}^\ell - p_i \right\|_2$, where $\mathbb{1}^\ell$ is an $k$ ($k$ being the number of classes) dimensional vector with $\left\| \mathbb{1}^\ell \right\|_1 = 1$ and the $\ell$'s position equals 1. When there is no training data, we compute the average of the nodes information ($y_i$) covered by scribbles of the same class (i.e., class $\ell$), and use this to represent class $\ell$ (denote as $Y_\ell$). Then $c_i^\ell = \left\| y_i - Y_\ell \right\|_2$. The regularization parameter $\lambda$ for all the ILP is set to 100, and the pairwise term $d_{ij} = e^{-\|y_i - y_j\|_2}$.

## A.5 VISUALIZATION OF EXPERIMENTS ON PASCAL VOC 2012

In this section, we present 2 experiments on Pascal VOC using our optimization algorithms, shown in Fig. 7. The figures are displayed in order of Table 4, where ILP-P-prob adopts the result of $\ell_0 H$-prob as its initial solution.

Table 4: Layout of the Pascal experimental images

|  | Left | Middle | Right |
|---|---|---|---|
| Top | Original image | $\ell_0 H$-RGB | $\ell_0 H$-layer 3 |
| Bottom | $\ell_0 H$-prob | MRF-prob | ILP-P-prob-5 |

In both experiments, we could see that using probability map is better than using low level feature maps. ILP-P-prob improves the result in the first one, while verifies the optimally in the second experiment, i.e., the initial solution of $\ell_0 H$-prob is already the global optimization for ILP-P-prob.

We observe that ILP-P-prob-5 has the same solution as MRF-prob in the first experiment, while improves MRF-prob in the second one.

## A.6 VISUALIZATION OF EXPERIMENTS ON CITYSCAPES

In this section, we present 3 experiments on Cityscapes using our optimization algorithms. The figures are displayed in order of Table 5, where "SEEG" and "Panoptic" denotes semantic and panoptic segmentation, respectively.

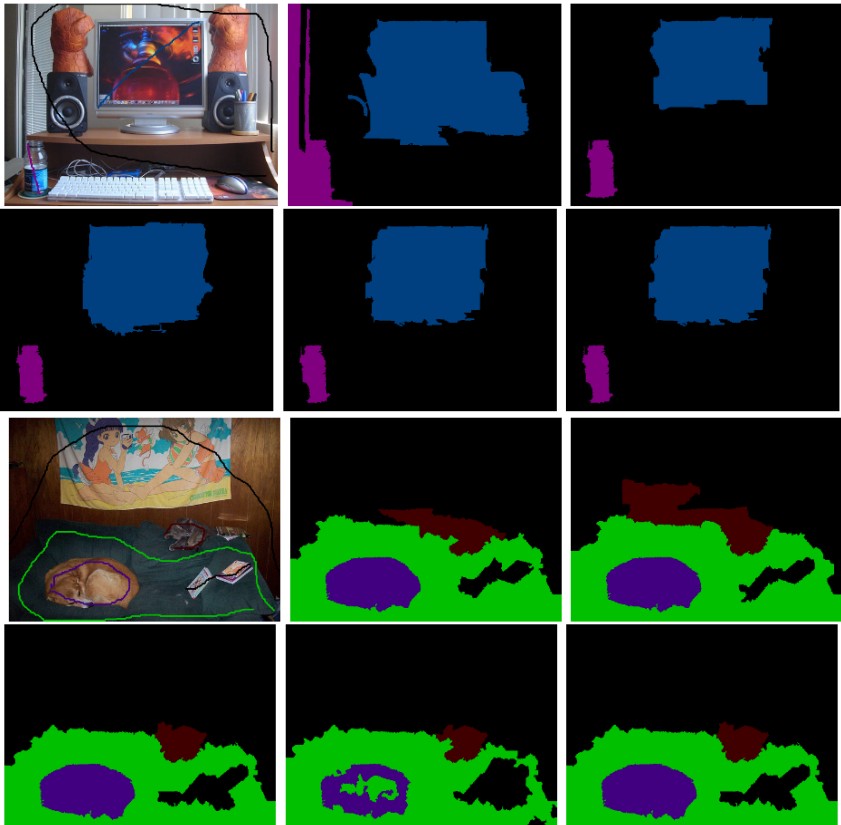

Figure 7: Visualization of 2 experiments on Pascal VOC validation set.

Supervised by scribbles, both semantic and panoptic segmentation have already good shape and sharp boundaries, even using just low level features of DCNN (pre-trained on other completely different datasets). The probability map from Cityscapes trained DCNN further improves the performance. But they are still limited by the quality of superpixels, especially for small and thin objects.

Table 5: Layout of the Cityscapes experimental images

|        | Left                   | Right                      |
|--------|------------------------|----------------------------|
| Top    | Original image         | Our artificial Scribbles   |
| Middle | $\ell_0 H$-layer 3 SSEG | $\ell_0 H$-layer 3 Panoptic |
| Bottom | $\ell_0 H$-prob SSEG    | $\ell_0 H$-prob Panoptic    |

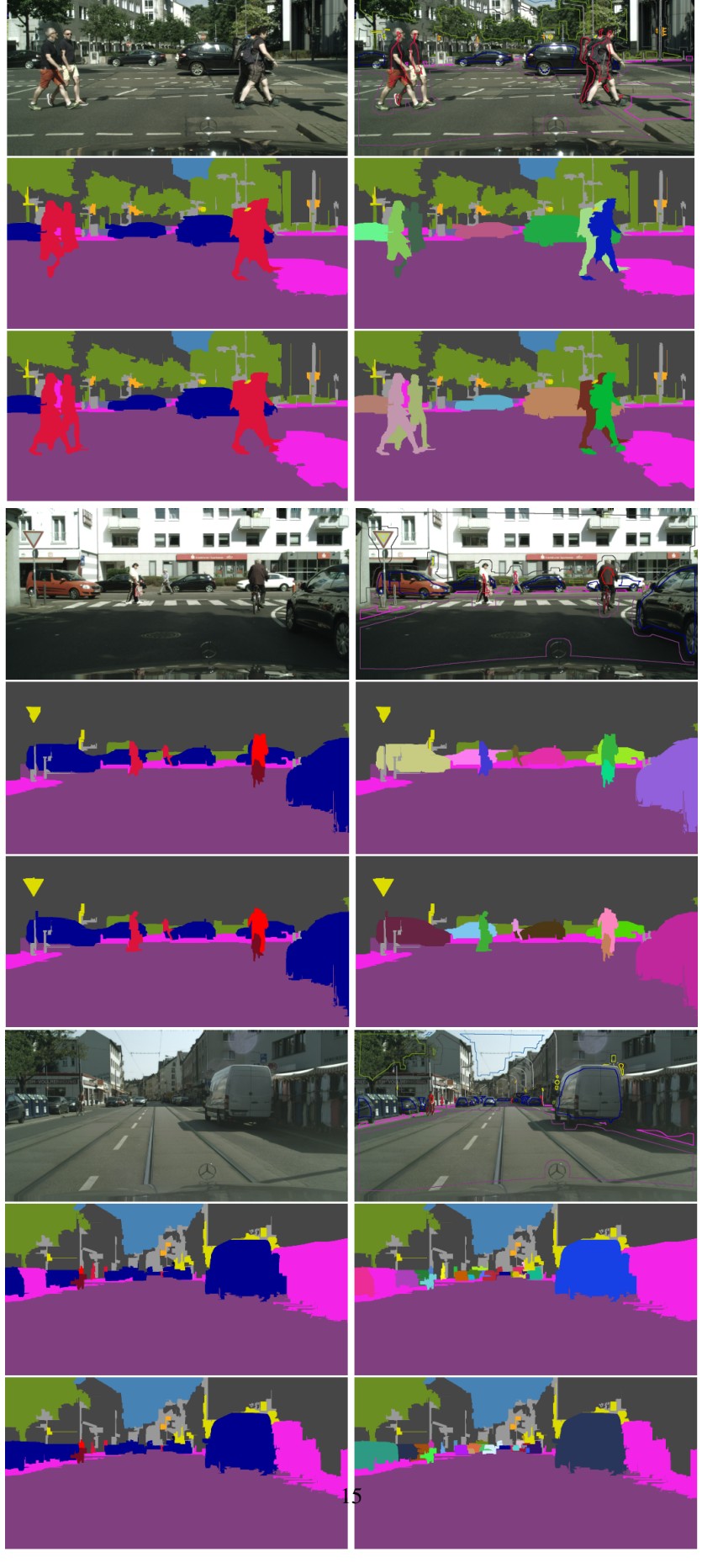

