# OpenReview forum: "Connectivity-constrained interactive annotations for panoptic segmentation"
_ICLR.cc/2020/Conference — Reject_

### Official Review · AnonReviewer3 · 2019-10-21
**Official Blind Review #3**

**Rating:** 1

**Review:**

Summary:
- key problem: efficiently leveraging scribbles as interactive supervision (at test time) for panoptic segmentation;
- contributions: 1) two algorithms leveraging scribbles via a superpixel connectivity constraint (one class-agnostic local diffusion heuristic, one class-aware with a MRF formulation), 2) experiments on PASCAL VOC 2012 and Cityscapes showing that both methods i) can achieve good performance without training data (using RGB values or pretrained features), ii) can improve the performance of a fully supervised baseline when using its probability maps as representation, and iii) can significantly improve performance beyond the state of the art on PASCAL when used interactively (90% mIoU with 3 rounds of corrective scribbles).

Recommendation: weak reject

Key reason 1: unclear novelty and relevance to ICLR.
- The paper proposes to apply two existing algorithms (Nguyen & Brown 2015, Rempfler et 2016) to a new task (interactive panoptic segmentation): what is the claimed novelty? What is specific to panoptic segmentation vs semantic or instance segmentation? Could the difference with related work in Section 2 be discussed more precisely?
- Furthermore, there seems to be no learning (representation or otherwise) involved in this submission. The paper mentions potential applications to weakly-supervised learning in Section 5, but it does not provide clear insights into what would be the benefits in terms of representation learning (vs. RGB, pre-trained features, or probability maps).
- Overall, this paper might be more tailored for a Computer Vision venue like CVPR.

Key reason 2: lack of sensitivity / robustness analysis.
- The scribbles are "simulated" using morphological operations on the ground truth (A.2, A.3): does this lead to realistic scribbles? Figure 3 (which is unclear) shows that the "scribbles" might be precise outlines or contours, which are very different than the expected scribbles illustrated in Figure 2. Contours provide much stronger information for segmentation, and are much more likely to effectively leverage the connectivity prior (esp. with the diffusion heuristic), but are they really scribbles / cheap supervision?
- What is the importance of the superpixel coverage by scribbles or missing scribbles or the location of scribbles relative to segment boundaries? What are the impact of realistic deviations from the expected scribble policy that are likely to happen in practice? Measuring sensitivity to different types of noise (by perturbing / dropping out scribbles) seems important to assess the practical usefulness and robustness of the method.
- PASCAL VOC and Cityscapes are small datasets. Experiments on bigger more recent ones like Mapillary Vistas and COCO are becoming the standard protocol in the instance/semantic/panoptic segmentation community. How would this method fare on those much more challenging datasets? What are the benefits of the proposed interactive methods in terms of scalability?

Additional Feedback:
- Fig. 4 is too low resolution / blurry;
- typos: "tarining set", "weekly supervised".

## Update following the rebuttal

Thanks to the authors for their replies. Sadly, my concerns are only answered at a high-level, and the consensus among reviewers is clear. Hence I confirm my rating to reject. I hope the feedback provided above will assist the authors in improving the work or finding a more suitable venue.

**Experience Assessment:**

I have published one or two papers in this area.

**Review Assessment: Checking Correctness Of Derivations And Theory:**

I assessed the sensibility of the derivations and theory.

**Review Assessment: Checking Correctness Of Experiments:**

I carefully checked the experiments.

**Review Assessment: Thoroughness In Paper Reading:**

I read the paper at least twice and used my best judgement in assessing the paper.

---

> ### Author Response · Authors · 2019-11-13
> **Response to reviewer 3**
>
> We thank the reviewer for the detailed comments.
>
> 1. Our method is not a direct apply of existing algorithms. The heuristic is greatly modified to comply with scribbles and in addition enforces connectivity of each scribbled region. Our ILP formulation extends previous MRF (only for class) to panoptic (both class and instance) by introducing dummy edge variables, and does not increase the complexity of the problem.
>
> 2. Although not a learning algorithm, ours are most suitable for annotating ground truth dataset, which is of fundamental importance to the data hungry deep learning method. Extension of our algorithms into the weakly/scribbles supervised learning framework similar to Lin et al (2016) can be a natural next step. In addition, we have conducted extensive experiments using RGB, lower level features and probability map as input to our algorithms. Results show that traditional machine learning algorithms can also benefit from deep learning by taking the its feature layers as input, which may be direction that is worth discovering.
>
> 3. Since the main application of our method is to annotate dataset, hence in a data annotation point of view, we argue that the artificial scribble is realistic. We tested drawing more strict (even closer to the boundary than the artificial ones) scribbles on Cityscapes, and it takes on average only 2 minutes per image, which is still a dramatic decrease in annotation time compared to 1.5 hours. On the other hand, by adopting the online available of VOC scribbles, we sort of already validate the robustness of our algorithms (84.6% mIoU and 90.6% after 3 correction scribbles).
>
> 4. We are re-running our algorithms that take pixels as input on Cityscapes. We will report the results once it’s done.
>
> 5. Our paper is mainly focused on the design of the algorithm and ILP formulation, hence we argue the experiments on two datasets suffice to validate the performance. In the future work when incorporating our algorithms into the weakly supervised learning framework, it is of great interest to test on more challenging datasets.

---

### Official Review · AnonReviewer1 · 2019-10-22
**Official Blind Review #1**

**Rating:** 3

**Review:**

This paper investigates scribble-based interactive semantic and panoptic segmentation.  The algorithms described build a graph on superpixels and do not require deep features or labels but rely on “scribbles” for supervision.  Given that semantic (and panoptic) annotation is very labor-intensive, advances in scribble-based annotation could significantly improve annotation time for new datasets; in applications where real-time performance is not required, scribble-based refinement of predictions could also be advantageous.

The experiments compare the proposed algorithms to deep baselines for VOC2012 and Cityscapes panoptic segmentation, and show impressive performance even without deep features. However they do not compare results to other scribble supervision methods to highlight the advantages of their approach over prior work.  I’d like for the experiments section to have a proper comparison to prior scribble algorithms (e.g. in section 4.4, comparing to other algorithms with the SOTA approach as baseline) to clearly show the advantage of their approach.

The results are impressive compared to the deep learning baseline, but I think further experimental validation should exist for properly comparing to prior work.

Post-rebuttal: I maintain my recommendation.

**Experience Assessment:**

I do not know much about this area.

**Review Assessment: Checking Correctness Of Derivations And Theory:**

N/A

**Review Assessment: Checking Correctness Of Experiments:**

I assessed the sensibility of the experiments.

**Review Assessment: Thoroughness In Paper Reading:**

I read the paper at least twice and used my best judgement in assessing the paper.

---

> ### Author Response · Authors · 2019-11-13
> **Response to reviewer 1**
>
> We thank the reviewer for the detailed comments.
>
> 1. Other scribble supervision method requires deep learning in the loop, while ours not. Since our contribution lies on the design of the algorithm/formulation that enforces connectivity, which could be served as a baseline for any weakly supervised learning method. Although we are pretty optimistic that adding the connectivity constraint would boost up the performance of Lin et al (2016), we leave the implementation and experiments of that as future work.

---

### Official Review · AnonReviewer4 · 2019-11-02
**Official Blind Review #4**

**Rating:** 3

**Review:**

This paper proposes two graph-based deep network feature fusion methods with connection constraints for semantic and panoptic segmentation. By incorporating additional scribble information to DCNN features, the methods yield improved results over the original network predictions on two popular semantic and panoptic segmentation datasets. Through interactively correcting error regions with the scribbles, further performance increase can be obtained.

I am not completely convinced by the novelty and experiments.
(1) First, the idea of smart annotation can be formalized as a weekly supervised segmentation problem where only part of the annotation is available. Can the authors justify how your work differs from those works solving the weekly supervised problem and what's your advantages. (Seed Expand and Constrain ... Alexander Kolesnikov; STC: A simple to Complex Framework for Weakly... Yunchao Wei; FickleNet Jungbeom Lee; etc..) Or, if possible, could you make a fair comparison with some existed weekly supervised approach on the final (semantic) result. Second, Potts model, MRF, K-nearest cut are known approaches. Thus I would like to know the deeper contribution of this work other than set constraints and solve ILP.
(2) The authors did not justify the use of less powerful models (DeepLabV2 and DRN) as both the inputs for l0H and ILP-P and the baseline comparison. The authors mentioned the current SOTA model (DeepLabV3+), which has achieved 79.55% mIoU on the CityScapes val set. However, they did not perform experiments using its probability map. It would be more convincing if the same performance gain can be achieved by using the SOTA model as inputs to the algorithms.
(3) The argument of achieving competitive results for panoptic segmentation is rather weak. To approach the panoptic segmentation problem, the authors essentially used scribbles to separate semantic region predictions into individual instances. Since the proposed algorithm requires as many scribbles to be drawn as there are regions, the baseline network only needs to predict semantic classes, and the algorithms uses the provided region IDs from the scribbles to segment individual instances. While this still has numerous applications in data annotation, it is somewhat unjust to claim that this method achieves competitive results in panoptic segmentation.
(4) The artificial scribbles for CityScapes experiments do not resemble human-drawn scribbles. Compared to the scribbles data for VOC12, the artificially generated scribbles for CityScapes experiments are visually idealistic. Rather than a stroke through the object, the generated is more similar to an outline of the object, which conveys a lot more information than a single line. Particularly when applied on super-pixels, it seems that super-pixels can easily be merged together by grouping any super-pixels within a scribble outline.
There are some other minor suggestions. For example, it might be clearer and easier to read if section 2.2.2 is presented in an algorithm format. Some minor typos and grammatical mistakes should also be corrected.


**Experience Assessment:**

I have published one or two papers in this area.

**Review Assessment: Checking Correctness Of Derivations And Theory:**

I carefully checked the derivations and theory.

**Review Assessment: Checking Correctness Of Experiments:**

I carefully checked the experiments.

**Review Assessment: Thoroughness In Paper Reading:**

I read the paper thoroughly.

---

> ### Author Response · Authors · 2019-11-13
> **Response to reviewer 4**
>
> We thank the reviewer for the detailed comments.
>
> 1. Our contribution is the design of a heuristic optimization algorithm and an ILP formulation that enforces connectivity (NP-hard) for interactive panoptic segmentation. Other than adopting lower level features of any deep learning basenet or its final probability map as input to our algorithms, no learning is evolved in our approach. Hence, it is not suitable to compare with other weakly supervised learning approach.
>
> 2. Since our contribution lies on the design of the algorithm/formulation, we did not focus on selecting and fine tunning the SOTA deep learning network. We searched and the networks presented in the paper are the best public available deep nets together with checkpoint on the internet for the time being.
>
> 3. We agree with the reviewer on this point, and that is why we added a condition on this statement, “given initial scribbles”.
>
> 4. Since the main application of our method is to annotate dataset, we argue that the artificial scribble is realistic for any data annotator. We tested drawing even more strict (closer to the boundary than the artificial ones) scribbles on Cityscapes, and it only takes on average 2 minutes per image, which is still a dramatic decrease in annotation time compared to 1.5 hours. On the other hand, by adopting the online available of VOC scribbles, we sort of already validate the robustness of our algorithms (84.6% mIoU and 90.6% after 3 correction scribbles).

---

### Official Review · AnonReviewer2 · 2019-11-03
**Official Blind Review #2**

**Rating:** 3

**Review:**

This paper introduces post-processing methods for panoptic (a combination of semantic and instance) segmentation, which are capable of using scribble annotations provided interactively by users. The proposed methods rely on (i) a discrete Potts model making use of RGB or DCNN features of the image, as well as the edge connectivity in a superpixel graph, and (ii) an integer linear program corresponding to a MRF with a pairwise data term. The proposed methods are evaluated on the Pascal VOC 2012 and Cityscapes datasets.

The paper is generally well written and easy to follow. The problem of panoptic segmentation is fundamental to computer vision, and as such, of relevance to the ICLR community. The proposed methods appear novel and worth pursuing.

A first reservation about the paper is that the method is primarily one of post-processing (after, e.g., extracting primary features from a DCNN), but the most common means of post-processing, namely conditional random fields, are not even mentioned, let alone compared against.

The other main reservation about the paper is that there are very few comparisons to the abundant literatures on either semantic or instance segmentation, and as such it is difficult to appreciate the paper’s contributions to these areas. Of note:

1. Evaluate on the COCO dataset, which is the current standard for segmentation ;
2. The scribble supervision method of Lin et al (2016) is mentioned, but not compared against.

Separately, the paper should compare the proposed method for semantic and instance segmentation with other methods that use weak-labels such as:
* Laradji, I. H., Vazquez, D., & Schmidt, M. (2019). Where are the Masks: Instance Segmentation with Image-level Supervision. arXiv preprint arXiv:1907.01430.
* Laradji, I. H., Rostamzadeh, N., Pinheiro, P. O., Vazquez, D., & Schmidt, M. (2019). Instance Segmentation with Point Supervision. arXiv preprint arXiv:1906.06392.
* Cholakkal, H., Sun, G., Khan, F. S., & Shao, L. (2019). Object counting and instance segmentation with image-level supervision. In Proceedings of the IEEE Conference on Computer Vision and Pattern Recognition (pp. 12397-12405).
* Zhou, Y., Zhu, Y., Ye, Q., Qiu, Q., & Jiao, J. (2018). Weakly supervised instance segmentation using class peak response. In Proceedings of the IEEE Conference on Computer Vision and Pattern Recognition (pp. 3791-3800).
* Zhu, Y., Zhou, Y., Ye, Q., Qiu, Q., & Jiao, J. (2017). Soft proposal networks for weakly supervised object localization. In Proceedings of the IEEE International Conference on Computer Vision (pp. 1841-1850).
* Ahn, J., & Kwak, S. (2018). Learning pixel-level semantic affinity with image-level supervision for weakly supervised semantic segmentation. In Proceedings of the IEEE Conference on Computer Vision and Pattern Recognition (pp. 4981-4990).

As the paper currently stands, given the gaps in the experimental evaluation, it is difficult to appreciate the contributions and complementarities of the proposed methods to the panoptic segmentation problem. As such, the paper would require more work before recommending acceptance at ICLR.

**Experience Assessment:**

I have read many papers in this area.

**Review Assessment: Checking Correctness Of Derivations And Theory:**

N/A

**Review Assessment: Checking Correctness Of Experiments:**

I assessed the sensibility of the experiments.

**Review Assessment: Thoroughness In Paper Reading:**

I made a quick assessment of this paper.

---

> ### Author Response · Authors · 2019-11-13
> **Response to reviewer 2**
>
> We thank the reviewer for the detailed comments.
>
> 1. We have compared ours to MRF shown in Table 2, and ours is 0.9% and 3.8% better in mIoU.
>
> 2. Our paper is mainly focused on the design of the algorithm and ILP formulation with connectivity constraints, hence we argue the experiments on the two datasets presented in the paper suffice to validate the performance. In the future work when incorporating our algorithms into the weakly supervised learning framework, it is of great interest to test on the more challenging datasets.
>
> 3. Our focus is the design of the two interactive optimization algorithm/formulation that enforces connectivity, which could be served as a baseline for any weakly supervised learning method. Although we are pretty optimistic that adding the connectivity constraint would boost up the performance of Lin et al (2016), we leave the implementation and experiments of that as future work.

---

### Author Response · Authors · 2019-11-13
**Response to all the reviewers**

First of all, we would like to thank all the reviewers for spending time reading our paper.
We have replaced Fig. 4 and corrected all typos mentioned by the reviewers.

We first want to remind the 4 highlights of our paper:
1.  We enforce the connectivity constraints (NP-hard) by introducing either a class-agnostic heuristic or a class-aware MRF integer programming, the later being an exact global optimization formulation.

2. Our method is not a learning method, but instead optimization algorithms that are suitable for inference based on RGB or lower level features input, or post-processing on existing learning algorithms (using their probability maps as input).

3. Ours does not necessarily require any available training data, i.e., it can use only the RGB, or the lower layer of any base network trained on arbitrary data set, as the input to our algorithms.

4. Compared to weakly supervised learning approach, the connectivity of scribbled region allow more control within the annotation framework (no outliers as shown in Fig. 1), hence is particularly suitable for annotating dataset.

Finally, the novelty of the paper comes from the re-design of the heuristic algorithm that complies with scribbles, and a novel ILP formulation that introduces dummy edge variables to deal with the multi-instance panoptic segmentation, while not increasing the complexity compared to the MRF for semantic segmentation. In addition, both methods enforce the connectivity prior.

---

### Decision · Program_Chairs · 2019-12-19

**Decision:**

Reject

**Comment:**

The paper proposes two methods for interactive panoptic segmentation (a combination of semantic and instance segmentation) that leverages scribbles as supervision during inference. Reviewers had concerns about the novelty of the paper as it applies existing algorithms for this task and limited empirical comparison with other methods. Reviewers also suggested that ICLR may not be a good fit for the paper and I encourage the authors to consider submitting to a vision oriented conference.